# Reliability of plastid and mitochondrial localisation prediction declines rapidly with the evolutionary distance to the training set increasing

**Sven B. Gould** \*, **Jonas Magiera, Carolina García García, Parth K. Raval** \*

Institute for Molecular Evolution, Heinrich–Heine–University Düsseldorf, Düsseldorf, Germany

\* gould@hhu.de; raval@hhu.de

## Abstract

Mitochondria and plastids import thousands of proteins. Their experimental localisation remains a frequent task, but can be resource-intensive and sometimes impossible. Hence, hundreds of studies make use of algorithms that predict a localisation based on a protein's sequence. Their reliability across evolutionary diverse species is unknown. Here, we evaluate the performance of common algorithms (TargetP, Localizer and WoLFPSORT) for four photosynthetic eukaryotes (*Arabidopsis thaliana*, *Zea mays*, *Physcomitrium patens*, and *Chlamydomonas reinhardtii*) for which experimental plastid and mitochondrial proteome data is available, and 171 eukaryotes using orthology inferences. The match between predictions and experimental data ranges from 75% to as low as 2%. Results worsen as the evolutionary distance between training and query species increases, especially for plant mitochondria for which performance borders on random sampling. Specificity, sensitivity and precision analyses highlight cross-organelle errors and uncover the evolutionary divergence of organelles as the main driver of current performance issues. The results encourage to train the next generation of neural networks on an evolutionary more diverse set of organelle proteins for optimizing performance and reliability.

## Author summary

Recent advancements in genome sequencing and machine learning have been instrumental in solving numerous biological challenges, such as the prediction of the folded state of proteins from sequences alone. An intriguing cell biological challenge is tracking the localization of proteins within cells. For example, some nuclear-encoded proteins localize to mitochondria, while some are sorted to plastids. Experimentally tracking the localization of each protein across thousands of species is laborious. Instead, researchers use machine learning algorithms to predict where proteins are likely to be localized based on their sequence. How reliable are these predictions? We evaluated the reliability of prediction tools across more than a hundred plant species. We found that, as the evolutionary distance between the species used for training the algorithms and those used for testing

---

---

**Data Availability Statement:** Supplementary figures are available in the supplementary information file. Additional supplementary data, in-house scripts and source data for the main and supplementary figures are available on Zenedo: https://zenodo.org/records/13924211.

**Funding:** We thank the Deutsche Forschungsgemeinschaft for grants awarded to SBG (SFB 1208-2672 05415 and SPP2237–440043394). The funders had no role in study design, data collection and analysis, decision to publish, or preparation of the manuscript.

**Competing interests:** The authors have declared that no competing interests exist.

increases, the accuracy of the predictions declines sharply. This perspective has allowed us to propose new strategies to improve these algorithms. We believe that training more distant plant genome sequences in combination with advances in artificial intelligence–and viewed through an evolutionary lens–will be crucial for developing localization prediction algorithms that are reliable across a wide range of species.

## Introduction

A plant encodes 20–30,000 proteins on average, of which many thousand are targeted to intracellular membrane bound compartments after or during translation [1–3]. The compartments owe their origins to bacterial ancestors directly or indirectly [4–12]. Mitochondria and plastids are of endosymbiotic origin and have transferred a majority of their coding capacity to the nuclear genome in the course of their transition from bacterium to organelle [13–15]. As a consequence, the vast majority of their proteins are translated in the cytosol and need to be imported. Protein translocation-related components of mitochondria such as TOM40, VDAC, TIM22, TIM23-PAM, OXA, SAM, HSP70, or the mitochondrial pre-sequence protease are likely of alphaproteobacterial origin [16–22], while many components of the plastid import machinery such as TOC75, OEP80, TIC20, the TAT pathway and several signal processing peptidases are of cyanobacterial origin [23–34]. Despite their evolutionary independent roots, the import machineries of mitochondria and plastids are united by principles of how they recognize the vast majority of their cargo.

Cytosolically-translated proteins destined for the mitochondrial matrix or the plastid stroma, thousands in sum, carry N-terminal targeting sequences (pNTS for plastid; mNTS for mitochondria) with many similarities and subtle differences. They concern the overall amino acid composition, processing peptidases and translocation motifs, and an overall charge difference among the more N-terminal region, in which mNTSs are enriched in arginine and pNTS are enriched in hydroxylated amino acids [35–39]. The subtle differences are still not fully understood, but determine whether a preprotein is targeted to mitochondria, plastids, or in the case of dual targeted proteins to both compartments simultaneously [40]. Considering the many remaining obstacles of *in vivo* protein localisation (time, resources, overexpression artefacts, impact of the tags on the cargo, or the simple unavailability of transfection methods for non-model systems) [41–47], hundreds of studies rely on algorithms that depend on the difference in NTS features for their localisation prediction. Furthermore, such prediction algorithms are integral parts of widely used databases such as Phytozome [48] or they are nested inside software packages such as InterProScan [49]. Hence, the algorithms are often used routinely, sometimes without a conscious decision to do so, and usually with a lack of knowledge on how reliable they work outside of the species on which they were trained.

*In-silico* localisation predictions from amino acid sequences were implemented concomitant with our understanding of cellular protein sorting [50–54]. Amino acid composition was used to differentiate between intracellular and secreted proteins [55–57], followed by the use of N-terminal features (e.g. charge and hydrophobicity) for signal sequence detection and cleavage site identification [52,58,59]. This channelled into early prediction algorithms such as PSORT [60] that relied on a relatively simple set of 'if and then' rules to predict signalling peptides and secreted proteins in Gram-negative bacteria and also included eukaryotes. PSORT II, an early formal expansion [61], incorporated a more sophisticated technique of k-nearest neighbours (kNN), which searches the query against a database of proteins with known localisations and assigns localisation of the nearest neighbours to the query. PSORTb [62,63]

introduced machine learning by including support vector machines for accumulating protein sequence features relevant to localisation. This culminated into WOLFPSORT (WPS from here on), one of the first sophisticated machine learning algorithms [64,65]. The algorithm uses approximately 20 features of the query sequence to calculate feature vectors, closest neighbours of which from the database are used for assigning a localization prediction. More than a decade later, the next generation of programs including Localizer and TargetP were released, which profited from more experimental data and advances in supervised machine learning[66,67]. Localizer is a classifier algorithm trained to differentiate between N-terminal regions of known organellar and non-organellar proteins. It abstracts 58 features of proteins from a positive and negative training set and the training process sets a boundary, which is a function of the weighed features. The features from a query are set on a hyperdimensional space and sorted into organelle or non-organelle using the boundary as a reference. TargetP 2.0 is an even more sophisticated algorithm that utilises bidirectional neural networks and multi-attention mechanisms on a network of interconnected, long short-term memory cells [66].

Apart from the training and sorting operations, the training datasets themselves also vary (Fig 1A). WPS for example used a database of 2004 (Uniprot v45.0), a time at which no genomes for bryophytes, ferns, let alone streptophyte algae or multiple organelle proteomes were available. Its training dataset was almost exclusively based on eudicot (for plastid) and animal (for mitochondria) sequences and the proteins were selected based on their annotation from the gene ontology database (GO; evidence codes: TAS, IDA, IMP; cut-off 12.4.2004). Two of these evidence codes (TAS and IMP) are indirect [67] and when used as a starting point, prone to multiplying errors. Localizer was trained on several hundred Viridiplantae organelle proteins from Uniprot (database until March 2016) and validated on the cropPal dataset (barley, wheat, rice, maize) as well as Uniprot Viridiplantae organelle proteins that were added between March and September of 2016. Of these Viridiplantae proteins, a vast majority was of eudicot origin. Tools such as cropPAL or SUBAcon (SUBcellular localisation database for Arabidopsis Consensus) significantly increase localisation prediction reliability, but only for selected eudicots on which they were optimized [43,68–70]. TargetP uses a relatively recent training data, including some green algal proteins, but again leaning heavily towards eudicots.

While vastly different in underlying algorithms and species data, TargetP, Localizer and WolFPSORT are among the algorithms with a superior reported accuracy. They are used abundantly across disciplines (Fig 1B), but are rarely benchmarked systematically across a wide range of species. Therefore, the impact of the skewed training on the performance and reliability of these algorithms outside angiosperms are unexplored. We made use of available, experimentally verified plant proteomes of mitochondria and plastids as well as protein clustering to investigate the reliability of these algorithms across species ranging from algae, across bryophytes and to angiosperms and organism with increasing research interest [71–77]. Our analysis brings forth deficiencies of these algorithms, caused by a combination of their inherent *modus operandi*, a lack of training on a diverse dataset, and the evolutionary dynamic nature of plant organelles [78]. Tracing the error sources allows to sketch an approach towards developing better algorithms that are capable of serving the diversity of the plant kingdom.

## Results

### Algorithms perform poorly outside of their training species

First, we compared the organelle proteomes predicted by the algorithms (the *in-silico* proteomes) with those of experimentally verified organelle proteomes (the *in-vivo* proteomes). Across species, *in-silico* proteomes comprise 3–15% of the proteins encoded by the genome of

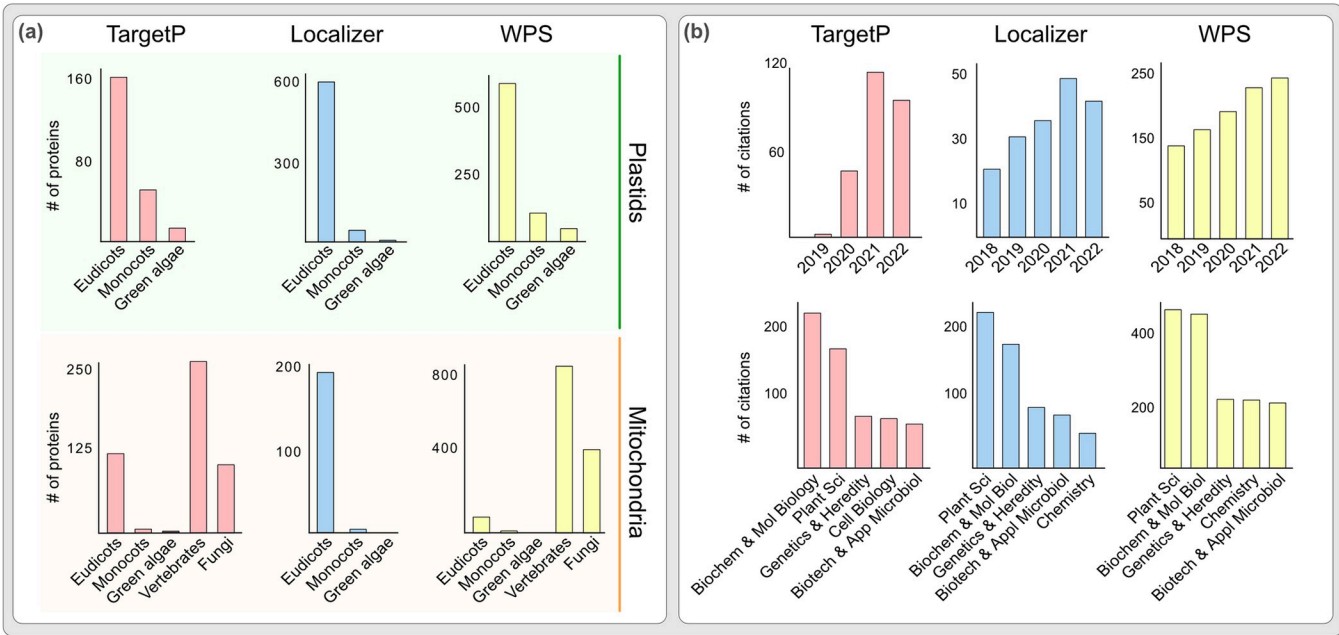

**Fig 1. Targeting prediction algorithms are frequently cited across disciplines and rely on a limited training set. (a)** Taxonomic distribution of plastid and mitochondrial training datasets used for the three commonly used predictions tools TargetP, Localizer and WoLF PSORT (WPS). **(b)** Distribution of citations across different disciplines for the three commonly used predictions tools TargetP, Localizer and WoLF PSORT (WPS) and for a time period ranging from 2018 until 2022. Numbers according to the Web of Science.

a given species, in contrast to the *in-vivo* numbers that usually range from 5–10% (S1 Fig). Overlaps between *in-silico* and *in-vivo* proteomes show a substantial false positive rate except for the *in-silico* plastid proteome predicted for *Arabidopsis* by TargetP (Fig 2A and 2B). Localizer and WPS show larger fractions of false positives than TargetP, especially for mitochondria (Fig 2B). The smallest overlap between *in-silico* and *in-vivo* proteomes are found for WPS. False negatives are generally predicted fewer on average than false positives, but still to a substantial number (Fig 2A and 2B). The sensitivity of TargetP and Localizer are similar, above 0.5 for plastid (i.e., correctly identifying more than half of the plastid proteins) and below 0.5 for mitochondria, whereas that of WPS is 0.3 or lower (Fig 2C and 2D). Since 2–5% of the proteins encoded in a nuclear genome have been localised to mitochondria or plastids *in-vivo* through proteomics or tagging (S1 Fig), a random sampling has a precision of 0.02–0.05; a perfect algorithm should have a precision of or close to 1. Between these two theoretical extremes, established algorithms currently perform closer to random sampling than to the best-case scenario, especially for mitochondria. The best improvement over a random prediction is observed for TargetP on *Arabidopsis* data, which however shifts ever closer to random the greater the evolutionary distance from *Arabidopsis* gets.

Combinations of algorithms reflect similar trends, where TargetP and Localizer together perform marginally better than each individually, as previously reported [79], albeit confined to the angiosperm plastid (Fig 2C). For mitochondria, the same combination captured less than 50% of verified proteins across species and any other combination captured less than 5% due the poor performance of WPS (Fig 2D). The precision was high in *Arabidopsis* for all combinations, too, but declined moving towards *Chlamydomonas* and regardless of combination (Fig 2C and 2D). To summarize, the predictions (for any individual algorithm or any combination) are more reliable for angiosperms and with a rapidly declining reliability with respect to algae and bryophytes (Fig 2).

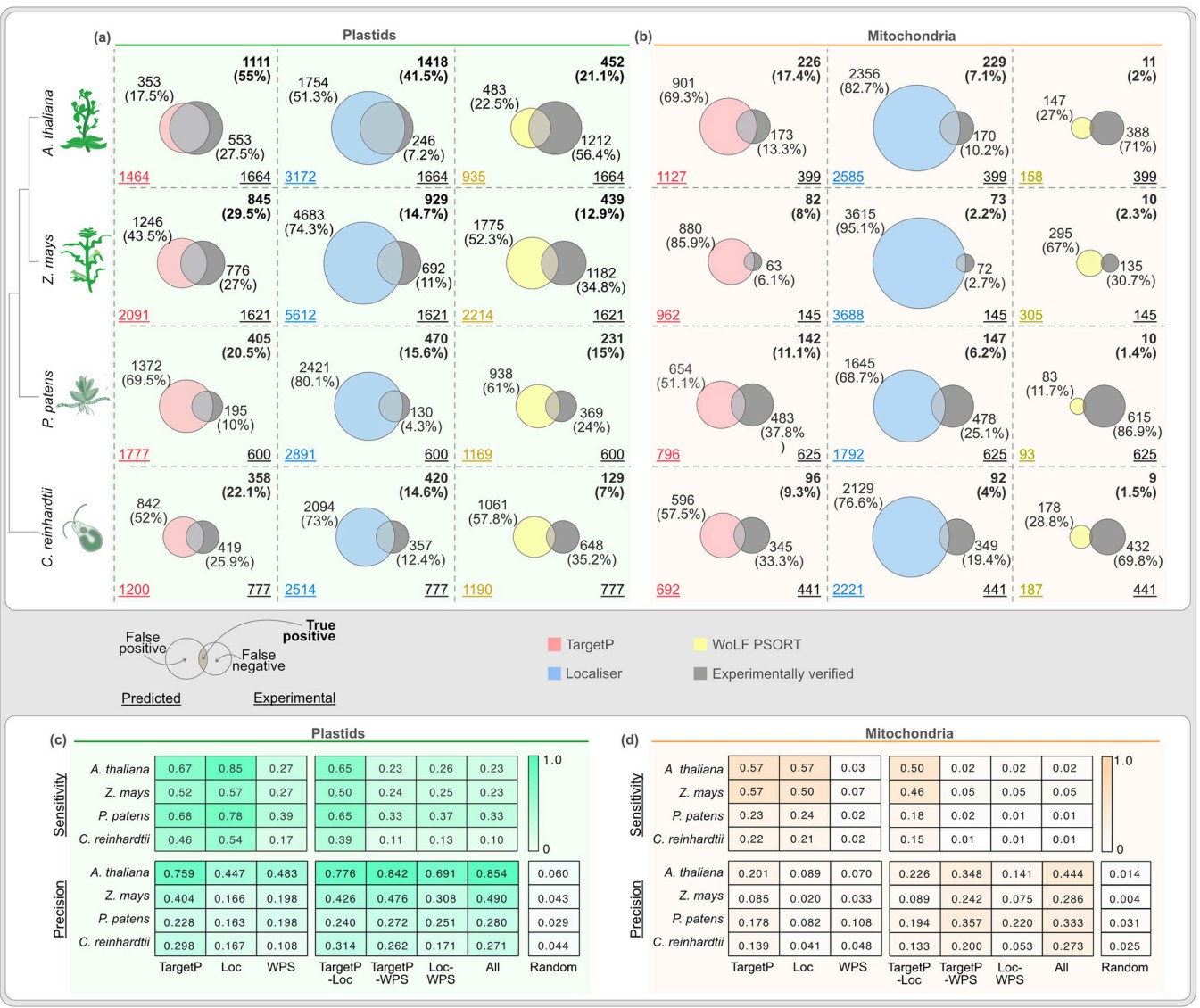

**Fig 2. Performance of algorithms outside the training species.** Comparison of predicted versus experimentally localised plastid **(a)** and mitochondrial **(b)** proteome numbers. Each Venn diagram of the top panel shows an overlap between predicted (left circles, colour-coded based on the algorithms used) and experimentally verified organelle proteomes (right circles, grey). The underscored numbers in the bottom corners show the total number of predicted (bottom left) and experimentally confirmed proteins (bottom right). The numbers of proteins that overlap (true positives) are provided in the top right corner in bold, while the numbers of non-overlapping false positives and negatives are shown next to each circle. See also the key for the Venn diagrams on the bottom left. Sensitivity, specificity and precision of individual algorithms and their combinations for plastid **(c)** and mitochondria **(d)**.

## Prediction performance declines as a function of evolutionary distance from the training data

To expand benchmarking across a larger evolutionary scale, we used TargetP (the best performing among the three) to predict organelle proteomes for 171 photosynthetic eukaryotes, for which genomes are available. Since organelle proteomes are scarce, we utilised orthology inferred organelle proteomes based on their sequence similarity with experimentally validated organelle proteomes [78]. Around 70–80% of predicted plastid proteins in eudicots could be validated by orthology based predictions (Fig 3 and S2 Table). Precision is

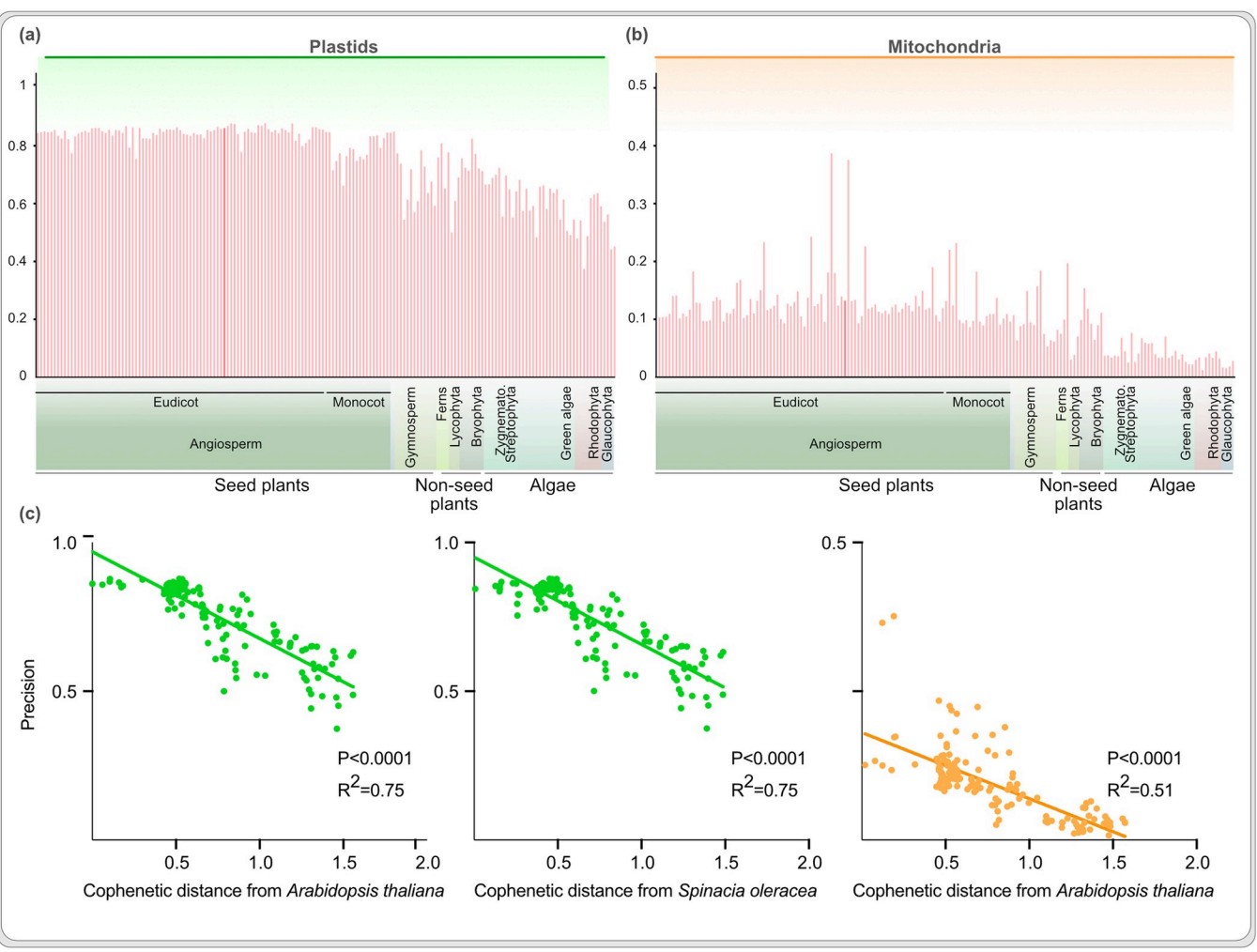

**Fig 3. Strong negative correlation between the precision of algorithms and the evolutionary distance from the training data.** Precision of TargetP across eukaryotes for plastid **(a)** and mitochondria **(b)**; *A. thaliana* is shown in darker shade. Taxonomic classification of test species is shown on the X axis, skewed towards eudicots due to genome sequence availability but similar to the training data (Figs 1A and S5). (c) Precision of TargetP as a function of evolutionary distance between the training species and 171 test genomes (plastid in green and mitochondria in orange).

lower for eudicot land plant sister lineages and algae, however, in accordance with patterns observed across the four species for which there is proteome data available (Fig 2). The precision for mitochondrial protein is lower, including for *Arabidopsis* (and related eudicots) and below 10% for algae. This large and diverse sample size allowed us to systematically and quantitatively test the impact of evolutionary distance between training and test species on the performance of the algorithms. TargetP is trained on 227 plastid and 499 mitochondrial proteins almost exclusively of eudicot or metazoan origin (S5 Fig). We calculated cophenetic (evolutionary) distances for each of the 171 test species from the most prominent training species, which reveals a significant negative corelation between the precision of algorithms across test species and the evolutionary distance of the test species from the training species. It underscores the need for an algorithm, whose performance is optimized with respect to evolutionary diversity. To this end, we next investigated sources of the prediction errors.

## The training bias of algorithms causes *in-silico* cross-organelle contamination

One likely source of false positives is the errors between the two organelles, caused also by the similarities in how their protein import machineries evolved. For example, a plastid protein can contaminate an *in-silico* mitochondrial proteome (Fig 4A) or vice versa (Fig 4B). Such errors can be quantified by overlapping the *in-vivo* proteome of one organelle with the *in-silico* proteome of the another: an overlap between the *in-vivo* plastid proteome and the *in-silico* mitochondrial proteome, highlights those plastid proteins that "contaminated" the *in-silico* mitochondrial proteome (Fig 4A). We observed that on average about a hundred or more plastid proteins were found across the four species in the *in-silico* mitochondrial proteomes (more frequently so with Localizer, in particular for the bryophyte and alga, S2 Fig) and a smaller number of mitochondrial proteins were identified in the *in-silico* plastid proteomes.

While NTSs of plastid and mitochondrial proteins share similarities, an mNTS contains a statistically significant higher net positive charge, while pNTSs contain a high number of serine and threonine residues among their first 20 amino acids [36]. It seems these differences became more pronounced later in plant evolution, since they are most striking in the angiosperms (Fig 4C and 4D, vertical green and orange lines). This is a good time to remember that more than 95% of discussed training datasets come from angiosperms (Fig 1A). Algorithms are inclined to sort NTSs based on these features and any NTS that deviates would be prone to an erroneous cross-organelle prediction, declining the performance of the algorithm. Indeed, NTSs of plastid proteins that showed a higher charge and/or a lower number of phosphorylatable amino acids than the average *Arabidopsis* pNTS, were predicted to be mitochondrial (Fig 4C) and NTSs of mitochondrial proteins that showed a lower charge and/or higher number of phosphorylatable amino acids than the average *Arabidopsis* mNTS were predicted to be plastid proteins (Fig 4D). These differences underscore that algorithms are trained to recognise and sort evolutionary late angiosperm targeting sequences, a bias that increases the error rate when facing proteins of algae and early branching plant species such as bryophytes.

The substantial number of cross-organelle prediction errors motivated us to investigate the predictability of proteins that are *in vivo* targeted to both, plastid and mitochondria. More than hundred such dually targeted proteins are identified in *Arabidopsis* [40], the plant proteomes of plastids and mitochondria corroborate such numbers and that is how we treated all proteins that overlapped in the proteome analyses. Algorithms can also predict the same protein to be plastid and mitochondria localised, either explicitly (by listing both these compartments) or implicitly (by providing similar probability scores for these two compartments). We considered such cases as predicted dual targeted proteins. *In-vivo* and *in-silico* dual targeted proteins hardly overlap, with hundreds of false positive and false negatives (Fig 4E). Except for maize, TargetP predicted most of the experimentally dual localized proteins (i.e. plastid and mitochondrion) to be only plastid localized or not to be organellar at all (Fig 4E and 4F). Localizer performed better than the other two with respect to quantity, but at the substantial cost of hundreds of false positives, and WPS failed to predict dual targeted proteins altogether. On the whole, all algorithms perform poorly on this task, sorting experimentally dual targeted proteins to only the plastid or no organelle at all, while also labelling non-organellar or plastid proteins falsely as being dual targeted likely as a result of cross-organelle errors (Figs 4A–4D and S2).

In summary, a combination of training bias and the evolution of targeting sequences ever since the origin of eukaryotes with mitochondria culminates into cross-organelle errors, which also affect the predictability of the dual targeted proteins.

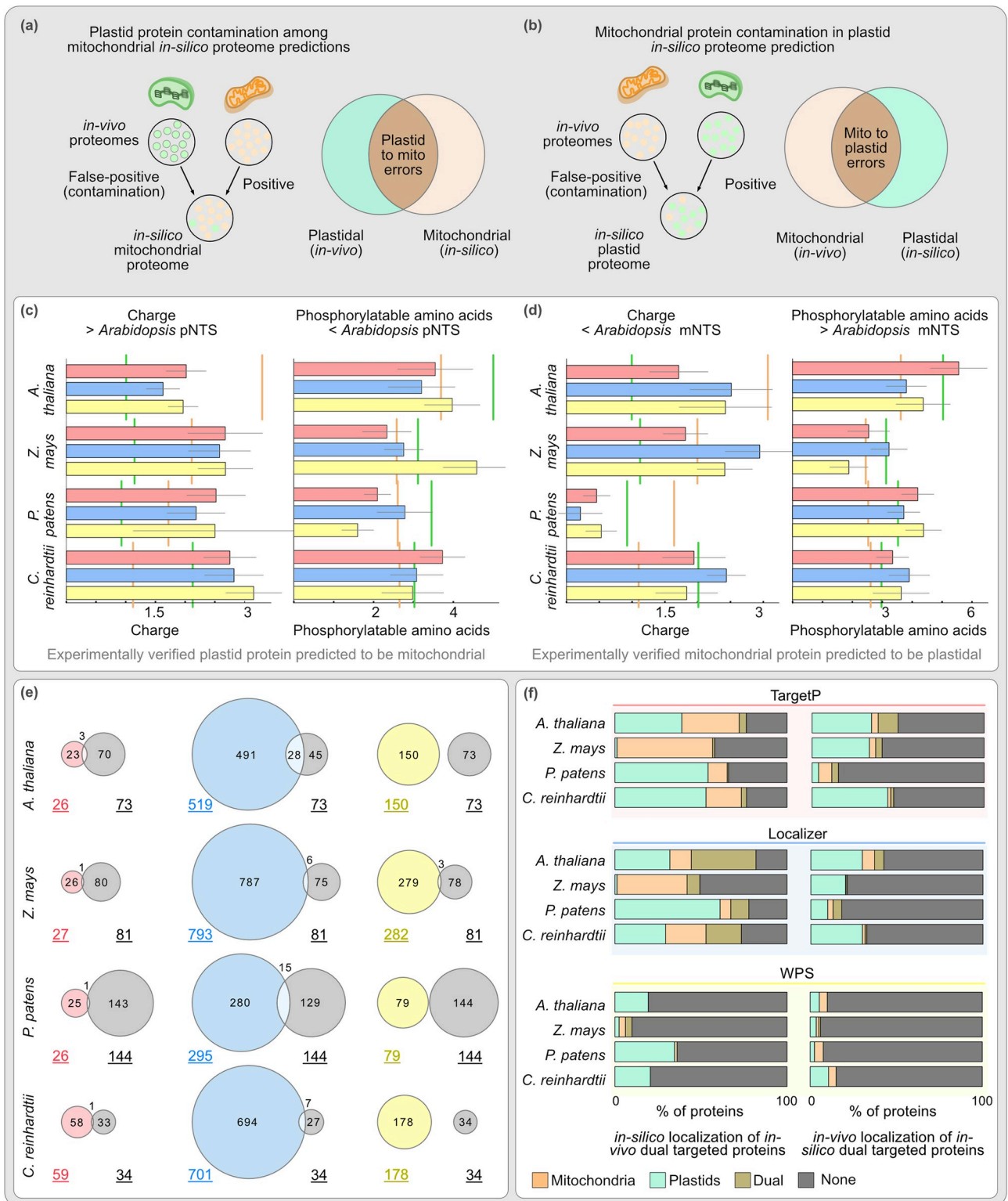

**Fig 4. Cross-organelle errors in proteome prediction due to physio-chemical properties of the NTS.** Cross organelle prediction errors could be either because an *in-vivo* plastid protein is *in-silico* mitochondria localised **(a)** or *vice versa* **(b)**. The overlaps between cross-organelle *in-vivo* and *in-silico* proteomes identifies these predictions errors. Analysis of the first 20 amino acids of pNTSs incorrectly predicted to be mitochondrial **(c)** and *vice versa* **(d)**. Average charge and phosphorylatable amino acids for NTS from all verified organelle proteins of each species are indicated by vertical green (pNTS) and orange (mNTS) lines. Error bars indicate standard error of mean (N = 4–331, S2 Fig). **(e)** Overlap between predicted (left) and

experimentally localised (right, in grey) dual targeted proteins. **(f)** Predicted (*in-silico)* intracellular localisation of experimentally verified (*in-vivo*) dual targeted proteins (left column) and experimentally verified (*in-vivo*) intracellular localisation of proteins that are predicted (*in-silico*) to be dual targeted (right column).

## Evolutionary dynamics and the diversity of organelles contribute to prediction inaccuracy

The endosymbiotic organelles of algae and plants have been co-evolving for over a billion-years and their proteomes continue to change and adapt [78,80,81]. During plant terrestrialization for instance, the plastid proteome of the algal ancestor expanded from a few hundred to that of the angiosperm plastid housing about 1500 proteins [78]. The algorithms predict there to be 1000 to 2000 plastid (and mitochondrial) organellar proteins even outside of angiosperms, 25% or less of which appear to be true positives (Fig 2). Together with the general pattern of the prediction performance worsening with the evolutionary distance to model angiosperms increasing, it prompted us to consider evolutionary dynamics of organelle proteomes as another error source.

We clustered all proteins from the four species into protein families [82], filtered the experimentally verified organelle protein families, and sorted them to be conserved (present in all four species) or to be unique (present in only one species) (S3 Fig and S1 Table). Around 150 protein families were found to be conserved across all proteomes, whereas a few hundred were unique. TargetP and Localizer missed around 30% of the conserved proteins, and WPS missed more (Fig 5A). For the unique plastid proteins, TargetP and Localizer performed well for *Arabidopsis* with declining success for the other species. WPS missed more than 75% of the unique proteins across the species (Fig 5A). For the conserved mitochondrial protein families, Localizer and TargetP predicted 50–70% correctly, whereas WPS missed more than 90% (Fig 5C). For mitochondria-unique proteins, the success rate ranged from 20–50% for Localizer and TargetP in *Arabidopsis* and other species, while WPS missed more than 90% across the species (Fig 5C). More than half of all protein missed out across the algorithms (i.e. false negatives of Fig 2), were present in only one of a given species (Fig 5B and 5d) and likely missed because of

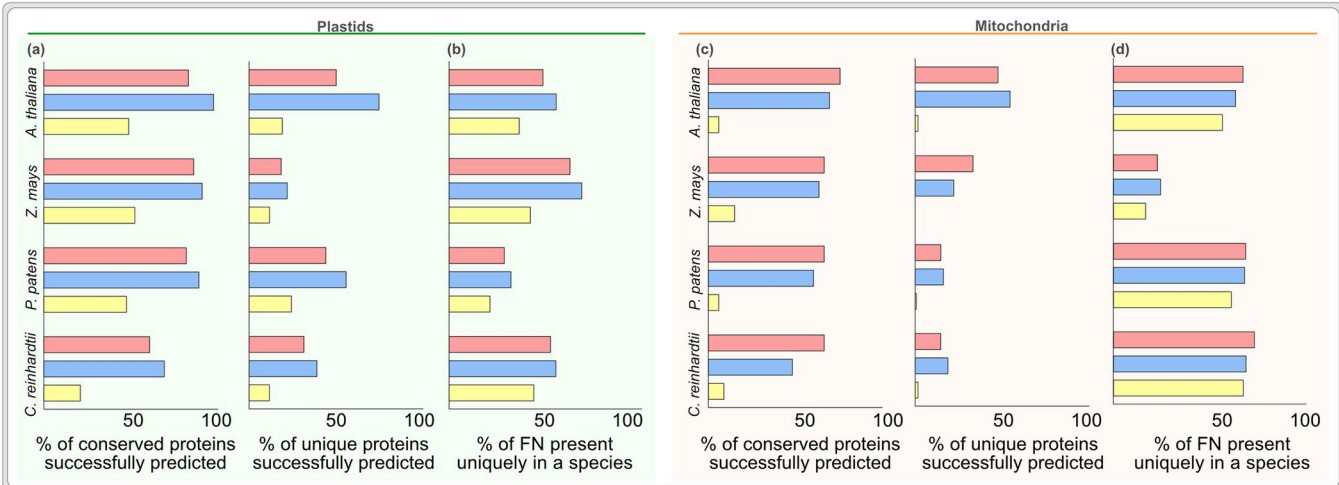

**Fig 5. Success rate of predicting unique versus conserved organelle proteins.** Success rate (sensitivity) of predicting experimentally verified conserved and unique proteins for **(a)** plastids and **(c)** mitochondria. All proteins from each species were sorted into conserved or unique based on sequence-based protein clustering (see methods, S3 Fig). Of the total plastid and mitochondrial false negatives (from Fig 2A and 2B), the number of proteins that were unique to a given species are shown for plastids (b) and mitochondria (d).

a lack of diverse training datasets. With the growing notion of organelle 'pan-proteomes', i.e. organelle proteins present in selected species or organelle sub-types [78,79,81,83–87], our analysis shows that algorithms are inadequate at capturing this pan-proteome nature or even the distant homologues of conserved proteins. To cover the species-specific organelle proteins and possibly the pan-proteome, future algorithms could be trained on missed proteins from across (Fig 2A and 2B) and from within species [88]. Algorithms could then sort predictions into a core- and pan-proteome, assigning credence and error margins likewise.

## Discussion

After its cytosolic translation, a plant protein needs to be targeted to the correct compartment if it is not to remain in the cytosol. Machine learning algorithms are used abundantly to determine where proteins are targeted, but they are trained on phylogenetically constrictive datasets (Fig 1A). They are often benchmarked on a limited number of species and, like the study at hand, identify TargetP among the most reliable prediction tools [68,69,79]. We benchmarked over a hundred photosynthetic eukaryotes including algae, using proteomes from four diverse representative species. The three widely used algorithms evaluated here perform poorly outside of model angiosperms, especially for mitochondrial cargo, for which the targeting prediction is only slightly better than random sampling. TargetP, the best performing among the three, has a fifty-fifty chance of sorting an algal plastid protein correctly and twice the chance of predicting a false positive. For mitochondria, the error margins are worse. For WPS, the most cited of the three analysed (Fig 1B), the chances of a wrong prediction are several times higher for plastid- and tens of times higher for mitochondrial proteins.

Such systematic error margins are a real issue, yet the output is trusted across individual studies directly (Fig 1B) or indirectly through being part of software packages and databases that cover genomes from hundreds of diverse species. Their genome annotations, however, receive localisation predictions from the same set of algorithms, but up to 70–80% of them can be wrong (Fig 3). While some of these 'false positives' can be attributed to experimental errors–the algorithms outperforming the experiments–this fraction is likely small. To reconcile experimental errors and contradictions [45–47], a combination of multiple experimental approaches under comprehensive projects such as SUBA or cropPAL [43,68–70,89] would be a first step to curate training data for algorithms incorporating chloro-, strepto- and bryophyte species (Fig 6). Algorithms thus trained on phylogenetically diverse datasets would improve reliability of large datasets, while being equally useful to diverse areas of fundamental and applied life sciences (Fig 1B).

Evaluating the error source in light of the cellular complexity and evolutionary cell biology of plants, allows to sketch improvement strategies for future algorithms. More than a billion years of co-evolution has resulted in plastid and mitochondrial proteomes and their import machinery, nuances of which affect the predictability of protein sorting. For instance, likely due to a selection pressure against plastid mistargeting, mitochondrial protein import evolved specific receptors such as TOM20 and TOM70 [90–94] that are unique to plant mitochondria. They have binding sites for cargo that differs from that of animal mitochondria [95–97]. Moreover, some of the predominant mNTS features from yeast (e.g. β-sheets) [98] are extremely rare in plant mNTSs and are rather similar to critical features of pNTS [99]. Therefore, including yeast mNTS in the training data sets generates false prediction to plastid in plants [100]. Such details are often not accounted for by the algorithms that are hitherto trained almost exclusively on animal and yeast sequences (Fig 1A). Consequently, algorithms require an upgrade to be able to predict plant mitochondrial proteomes and training them on plant mitochondrial proteins, and accounting for the receptor platform differences, is essential (Fig 6).

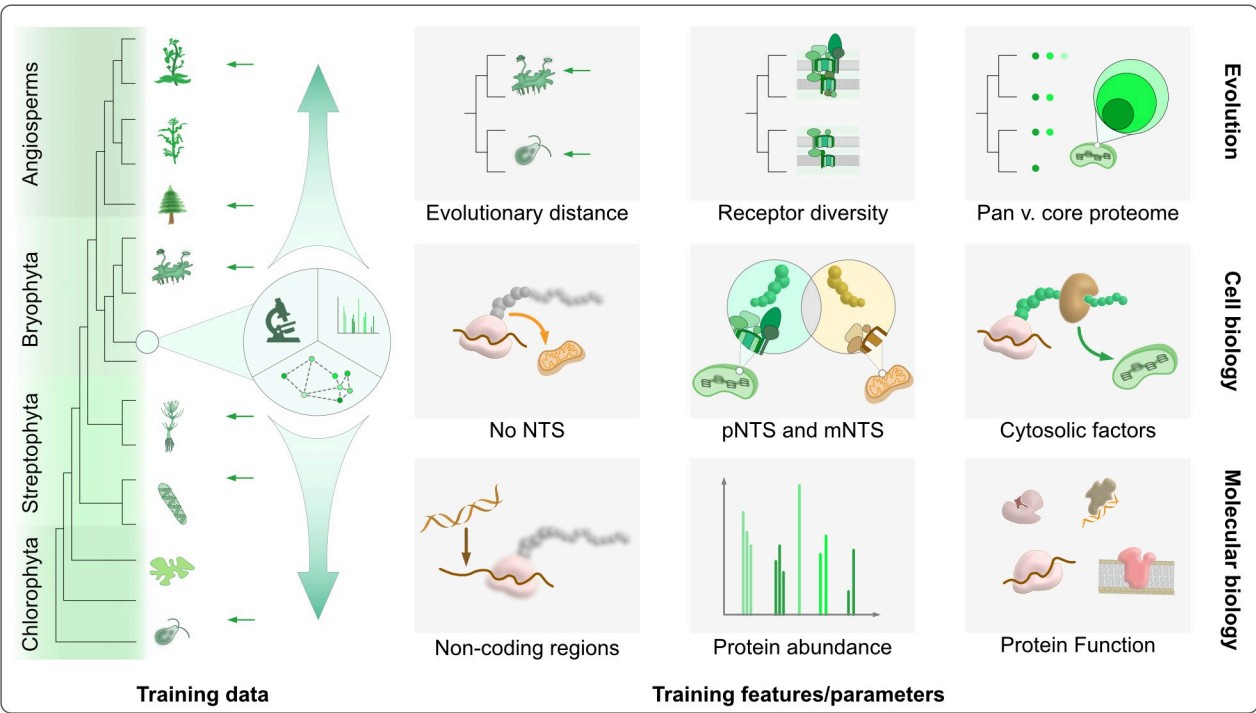

**Fig 6. A framework for improving localisation prediction algorithms.** Strategies to improve prediction reliability involve changes in the curation of the training data as well as the training procedure. The training data should ideally be collected from a range of diverse species and for each be based on different experimental techniques that support a training protein's localisation (e.g. reporters, mass spectrometry, coexpression, interactions). Proteins with non-canonical internal motifs, or those dually targeted need to be taken into account (as they help to better distinguish between pNTS and mNTS features) and validated data could be sorted according to whether it is part of a core- or pan-proteome. Classifiers on which the algorithms are trained could include parameters such as the evolutionary distance of a species, non-coding regions, or a protein's abundance as a currently neglected factor. One can expect that the combination of multi-dimensional parameters from evolutionary biology, cell biology and molecular biology on evolutionary diverse species will significantly improve the next generation of machine leaning algorithms that serve localisation (and function) predictions.

The impact of organelle co-evolution appears to be more pronounced in angiosperms sequences (the training dataset), which evolved features different from other clades, such as longer pNTSs and different physicochemical properties of NTSs in general [35,101–103]. The details of NTSs are mostly studied in a few angiosperms [104–108], however, and in league with the skewed training (Fig 1A) compromises the performance of algorithms outside of angiosperms. For instance, we utilised BaCeLlo [109], trained on *M. musculus*, *S. cerevisiae*, *C. elegance*, and *A. thaliana*, on false negatives (i.e. organelle proteins missed by each of the three algorithm) from *Physcomitrium*. It sorted up to 50% of them to mitochondria, regardless of their experimental localisation (S6 Fig). This further underscores that an algorithm trained on more than one species can also perform poorly outside of angiosperms, when the training focusses on animal and angiosperm sequences alone. A better understanding of NTSs outside of angiosperms remains a bottleneck for developing better algorithms, as much as it remains an unchartered territory in the field of protein import evolution.

Some NTSs are ambiguous and identified equally well by the import machineries of mitochondria and plastids. Although these dual targeted proteins are small in number, they play a key role in information processing [110,111] and have been theorized to reroute whole metabolic pathways [112]. The process of dual targeting appears to be conserved [113,114], rarely lost [113] and can arise by small changes in the NTS [115]. Therefore, it is likely to be common across species, but outside of the model systems the identification of dually targeted proteins is

limited. Algorithms are currently of little use in this respect, as they assign dual targeted proteins usually only to the plastid, sort plastid proteins to mitochondria as reported previously [116], or falsely predict many sole plastid proteins to be dually localised. *In vitro* protein import assays with purified organelles also localise many plastid proteins to plastid and mitochondria both, which complicates the matter [117–120]. Ambiguous Targeting Predictor (ATP), an early algorithm tailored towards dual targeting [121], predicted ca. 500 *Arabidopsis* proteins to be dual targeted, of which only 30 have been experimentally verified to date (S7 Fig). CropPAL [68] predicts several hundreds to a few thousand dually targeted proteins for six species, of which <5% are experimentally supported by the same database (S8 Fig). In a previous study, SUBAcon predicted around 30 proteins to localise to both mitochondria and plastids [69], a performance comparable to that of TargetP (Fig 2A). This further underscores that algorithms either under- or massively overpredict dual targeted proteins and this remains a crucial challenge. Such *in vitro* and *in silico* errors limit our understanding of in vivo dual targeting mechanisms. Studying protein dual targeting outside of the established model systems would elucidate general strategies of dual targeting. In the interim, explicitly training algorithms on verified dual targeted proteins could help to identify targets for experimental investigation.

Our analysis also shows that prediction reliability at large declines significantly, when phylogenetically diverse species come into play. In contrast to previous benchmarks, we systematically quantify the extent of error, as test data diverge from training data using phylogenetic distance. Such quantifications allow algorithms to provide a confidence interval–a feature largely missing–based on the evolutionary distance between the training and test data. They can also be used to reject a query, if the evolutionary distance value crosses a certain threshold. The next step could be to systematically use evolutionary distance as a parameter in machine learning, weight of which can be assigned during training-testing iterations on mitochondrial and plastid proteomes from diverse species. When doing so, and in the absence of proteome data, one could commence with canonical and universally accepted organellar marker proteins. Lastly, most algorithms assume the presence of an NTS and attempt to sort a query to organelles. N-terminal targeting peptide-independent import, however, is known and the nature of cargo recognition often more involved [122]. This presents another source of error and requires to predict a localisation on classifiers independent of targeting sequence features alone and they could include e.g. homology, GO or KEGG annotations, or even promoter length [123].

Apart from prediction errors, organelle proteomes can vary also across closely related sister species. For example, a systematic comparison of organelle proteomes between several eudicots and monocots showed that proteomes within crops were more similar to each other than to *Arabidopsis*, highlighting the clade-specific nature of organelle proteomes [70]. This study also highlighted that functions influence how conserved the localisation is across species (and that e.g. the localisation of proteins involved in metabolism are less conserved). Such examples motivates algorithms tailored to a given species [123,124] or clade [116,125], but recent computational power and AI advances encourage us to try the opposite and attempt to develop more generalised algorithms, which abstract clade-specific peculiarities. Moreover, not all proteins are equally abundant in organelles, but they often contribute equally to the training process of algorithms. It is conceivable that NTSs have evolved differences based on protein abundance. Inclusion of relative abundance of proteins in the training process might improve the predictions and reveal novel strategies of protein sorting. Applying such strategies to train algorithms on a diverse set of species (Fig 6) would increase their generalisability.

In conclusion, as advances in in proteomics [126,127], genomics [75,77,128–132], and machine learning [133,134] set a stage for future prediction algorithms, our analysis serves as a

reminder that considering evolutionary diversity is key, also to a better modelling of protein sorting.

## Methods

### Algorithms

All algorithms were installed on a local server supported by the ZIM at the HHU Düsseldorf. Full proteomes were analyzed using TargetP 2.0 (https://services.healthtech.dtu.dk/services/TargetP-2.0/) with the setting 'pl' (plant derived); with Localizer 1.0.4 (https://localizer.csiro.au/software.html) with Python 2.7 and setting '-p'; WPS 0.2 (https://github.com/fmaguire/WoLFPSort) with setting 'plant'. The outputs were processed using the script 'Algorithm_predicted_proteins.py' and 'batch_process_targetP.py' (for targetP across eukaryotes in Fig 3). The dual targeted proteins were retrieved using the script 'dual_targeting_prediction.py'. The number of citations for each algorithm were retrieved from the Web of Science.

### Source genomes and organelle proteomes

Genomes of all chloroplastida species were downloaded from Kyoto Encyclopedia of Genes and Genomes (KEGG) [135]. Experimental organelle proteomes were retrieved from published literature and database as follows: *Chlamydomonas reinhardtii* (chlorophyte algae) [136,137], *Physcomitrium patens* (bryophyte) [138], *Zea mays* (monocot) [42], *Arabidopsis thaliana* (eudicot) [42]

### Evaluation of algorithms

We evaluated the performance in species from four diverse chloroplastida species. A protein present in verified proteome and absent in prediction was categorised as false negative. A protein absent in verified proteome and present in prediction was categorised as false positive. A protein present in both, verified and experimental, proteome was categorised as true positive. Sensitivity (i.e. true positive rate) was calculated as a ratio of true positive and true positive + false negative.

$$\text{Sensitivity} = \frac{TP}{TP + FN}$$

Precision was calculated as a ratio of true positive and all predictions.

$$\text{Precision} = \frac{TP}{TP + FP}$$

The script 'Sensitivity_precision.py' was used for the calculation of sensitivity and precision and the Venn diagrams were generated using 'Venn_diagrams.py'.

For a combinatorial approach, organelle proteomes were predicted individual by each algorithm and proteins present in the prediction of both or all three algorithms were filtered for further evaluation against experimental proteome. TargetP2.0 predicted 'thylakoid' proteins as a category distinct from 'chloroplast' and therefore around 100 thylakoid proteins were not included under 'chloroplast predicted' category. Inclusion of these proteins do not change broad patterns by more than a few percentage (S4 Fig, as compared to Fig 1A).

### Protein family clustering and annotation

Whole proteomes of all species were clustered into protein families using Orthofinder version 2.5.4 [82]. Source genomes of all species was taken from KEGG [135].

### Analysis of N-terminal targeting sequences and prediction of the dual targeted proteins

The first 20 amino acids of each protein were retrieved from the whole genome assemblies using the script 'get_first_20AA.py'. Charge was determined by assigning -1 to D,E; +1 to K,R; +0.5 to H and 0 to the rest of the amino acids. The total number of serine and threonine were counted as phosphorylatable amino acids. Both these features were retrieved using the script 'charge_phospho.py'. The verified dual targeted proteins were inferred from overlapping the experimental proteomes of mitochondria and plastid for each species. TargetP sorts proteins to only one intracellular localisation, which gets the highest probability. However, if probability of mitochondria and plastid both were above 0.35, we considered that protein to be dually targeted. WPS and Localizer predicted more than one localization explicitly, and hence proteins predicted as plastid and mitochondria, were labelled dually targeted. This was done using the script 'dual_targeting_prediction.py'. The experimental localisation of predicted dual targeted proteins and the predicted localisation of experimentally dual targeted proteins were received by the scripts 'Exp_dual_insilico_Loc.py' and 'Exp_loc_of_predicted_DTP.py'.

### Corelation between evolutionary distance between training and test species and precision

The genomes of 202 eukaryotes and its phylogeny, along with inferred plastid and mitochondria localisation protein families were retrieved from previous study [78]. TargetP analysis and calculation of prevision was performed as described above. The protein identification numbers of the training data and their species details were retrieved from TaregtP 2.0 (https://services.healthtech.dtu.dk/services/TargetP-2.0/) and uniprot (http://uniprot.org/) websites respectively. The frequency at which different species were represented in the training data was calculated and top two plant species were chosen for evolutionary distance analysis. Their phylogenetic distance from each of the test species was calculated in rstudio using APE [139] and the script 'cophenetic_distance.r'. The precision v.s. evolutionary distance plots and linear regression were conducted in Graphpad prism.

## Supporting information

**S1 Fig. Experimentally verified and predicted organelle proteins as a percentage of the whole genome.** Proteomes of each species from KEGG (Kyoto Encyclopedia of Genes and Genomes) were used as an input for the three algorithm to get proteins predicted as plastid and mitochondria. Their experimental proteomes were taken from organelle proteome databases and literature (see methods). Predicted and experimentally verified plastid (on the left) and mitochondrial (on the right) proteins were plotted as a percentage of all proteins encoded by a given species.
(TIFF)

**S2 Fig. Number of proteins predicted between plastids and mitochondria.** The number of experimentally verified plastid proteins that got predicted as mitochondrial proteins by the three algorithms (on the left) and experimentally verified mitochondrial proteins that got predicted as plastid proteins (on the right).
(TIFF)

**S3 Fig. Protein clustering and filtering of organelle protein families.** All proteins from the four photosynthetic eukaryotes and sorting of protein clusters into plastid (on the left) and mitochondrial family (on the right). Each circle is a protein from a species. In the first step

(shown on top), source protein sequences from available species were clustered into protein families (shown at the bottom). If a protein family consisted of an experimentally verified plastid protein (in green, on the left) or a mitochondrial protein (in orange, on the right), the protein family was sorted as a plastid or mitochondrial protein family.
(TIFF)

**S4 Fig. Chloroplast predicted proteins from TargetP with thylakoid predictions included.** Comparison of chloroplast+thylakoid proteins predicted by TargetP2.0 with experimentally localised proteins across species. Each Venn diagram represent data similar to that of Fig 1A, expect now supplemented with 'thylakoid' predicted proteins under the category 'plastid'. The Ven diagrams show an overlap between predicted (left circles) and experimentally verified organelle proteomes (right circles, grey). The underscored numbers in the bottom corners show the total number of predicted (bottom left) and experimentally confirmed proteins (bottom right). The numbers of proteins that overlap (true positives) are provided in the top right corner in bold, while the numbers of non-overlapping ones (false positives) are shown next to each circle. See also the key for the Venn diagrams on the bottom right.
(TIFF)

**S5 Fig. Taxonomic distribution of TargetP 2.0 training dataset.** The targetP2.0 training proteins were downloaded from the original publication and based on their swissprot IDs, their full taxonomy was recovered and number of training proteins per species is plotted here for plastid (a) and mitochondria (b) (with species color coded as per their taxonomy, taxonomy class 'others' include: protozoa, insect, nematode, fish, amphibian, amoebozoa, dinoflagellate).
(TIFF)

**S6 Fig. *Physcomitrium* false negative sorted by BaCeLlo.** Experimentally verified *Physcomitrium* organelle proteins that were missed by each algorithm (i.e. the false negative) were used as queries to BaCeLlo to check whether it can sort them correctly. BaCeLlo sorted ca. 50% of them to mitochondria or cytosol, regardless of their verified locations, showing overall affinity for mitochondrial sorting and a lack of reignition for targeting sequence.
(TIFF)

**S7 Fig. Validation of *Arabidopsis* dual targeting proteins predicted by Ambiguous Targeting Predictor.** Dually targeted proteins predicted in *Arabidopsis* by the Ambiguous Targeting Predictor (ATP) compared with mass-spec confirmed dual targeted proteins from *Arabidopsis* shows that ATP missed more than half of *Arabidopsis* dual targeted proteins and predicted ten times more proteins to be dually targeted.
(TIFF)

**S8 Fig. Dual targeting predictions by cropPAL.** cropPAL incorporates a dozen algorithms of which if the majority of algorithm sorts a protein to plastid and mitochondria both, we consider it to be predicted dual targeted. For a given protein, if at least one experimental study experimentally showed plastid and mitochondrial localization, we consider it to be experimentally verified dual targeted protein. If more than one study converge onto plastid and mitochondria, cropPAL labels it as 'experimental consensus'. Overlap of the three categories (predicted, experimentally verified and experimental consensus) is shown for six species, and they generally underscore overprediction of dual targeting.
(TIFF)

**S1 Table. List of protein families across species.**
(XLSX)

**S2 Table. List of predicted organelle proteins by targetP and orthology approach.**
(XLSX)

**S1 Data. Source data for Fig 1A and 1B.**
(XLSX)

**S2 Data. Source data for Fig 2A–2D.**
(XLSX)

**S3 Data. Source data for Fig 3A–3C.**
(XLSX)

**S4 Data. Source data for Fig 4C–4F.**
(XLSX)

**S5 Data. Source data for Fig 5A–5D.**
(XLSX)

**S6 Data. Source data for S1 Fig.**
(XLSX)

**S7 Data. Source data for S2 Fig.**
(XLSX)

**S8 Data. Source data for S4 Fig.**
(XLSX)

**S9 Data. Source data for S5A and S5B Fig.**
(XLSX)

**S10 Data. Source data for S6 Fig.**
(XLSX)

**S11 Data. Source data for S7 Fig.**
(XLSX)

**S12 Data. Source data for S8 Fig.**
(XLSX)

## Acknowledgments

We acknowledge support from the high-performance computing cluster (HILBERT; ZIM at the HHU Düsseldorf) and Michael R. Knopp from the Heinrich–Heine–University Düsseldorf. We also thank William Martin for a discussion on including protein quantity as a parameter.

## Author Contributions

**Conceptualization:** Sven B. Gould, Parth K. Raval.

**Data curation:** Jonas Magiera, Parth K. Raval.

**Formal analysis:** Jonas Magiera, Parth K. Raval.

**Funding acquisition:** Sven B. Gould.

**Investigation:** Jonas Magiera, Parth K. Raval.

**Methodology:** Parth K. Raval.

**Project administration:** Sven B. Gould.

**Software:** Jonas Magiera, Parth K. Raval.

**Supervision:** Parth K. Raval.

**Validation:** Parth K. Raval.

**Visualization:** Parth K. Raval.

**Writing – original draft:** Sven B. Gould, Carolina García García, Parth K. Raval.

**Writing – review & editing:** Sven B. Gould, Carolina García García, Parth K. Raval.

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
