## [Decision Letter · Decision Letter 0]

12 Jul 2024

Dear Dr. Gould,

Thank you very much for submitting your manuscript "Performance of localization prediction algorithms decreases rapidly with the evolutionary distance to the training set increasing" for consideration at PLOS Computational Biology.

As with all papers reviewed by the journal, your manuscript was reviewed by members of the editorial board and by several independent reviewers. In light of the reviews (below this email), we would like to invite the resubmission of a significantly-revised version that takes into account the reviewers' comments.

We cannot make any decision about publication until we have seen the revised manuscript and your response to the reviewers' comments. Your revised manuscript is also likely to be sent to reviewers for further evaluation.

Sincerely,

Anders Wallqvist

Academic Editor

PLOS Computational Biology

Daniel Beard

Section Editor

PLOS Computational Biology

Reviewer's Responses to Questions

**Comments to the Authors:**

Reviewer #1: In this manuscript, the authors reported their benchmark study on the prediction accuracy of three widely used algorithms for localization into plant (or photosynthetic eukaryotes’) plastids and mitochondria using experimentally determined proteome data supplemented by homology-based data. Their results show that the performance of these tools is not practically reliable and that the accuracy becomes even worse in plant species with more evolutionary distances from (well-studied) eudicots or angiosperms. Overall, their messages are clear; this work will be an important warning for end users and encourage the developer of prediction algorithms to address this issue. But the following points should be improved before its acceptance:

1. Although the claim in the title is likely correct, they should claim only what they showed in this study.

2. The authors seem to presume that all nuclear-encoded plastid/mitochondrial proteins carry N-terminal targeting sequences. However, this assumption has not been proven (if I understand correctly). If this assumption is incorrect, we cannot expect TargetP to be 100% accurate because it only predicts the existence/absence of N-terminal signals. This point should be noted clearly.

3. More importantly, their main message, “Performance of localization prediction algorithms decreases rapidly with the evolutionary distance …,” seems to imply the existence of a general/ideal predictor applicable to the sequences of any species. However, as the authors point out, the translocation complexes can vary between species. The efficiency of recognition of a typical signal in one species can be greatly different in another species. Therefore, the only reasonable solution seems to prepare specific predictor for each evolutionary group of species. Using machine learning techniques, such a remedy would be rather trivial (if there is enough data size).

4. In this sense, the authors should show how different the N-terminal signals are between different evolutionary groups (in the case of mitochondria, animal sequences should also be compared) (like Fig.4c & d).

5. Although this does not matter for users, Wolf PSORT was released 12 years before TargetP v.2.0. Obviously, there was not much sample data for various species at that time. So, at least from the side of algorithm developers, the comparison ignoring this point does not make much sense.

6. The current explanation of the Localizer algorithm should be improved. The first version of PSORT also could treat eukaryotic sequences (PMD 1478671).

Reviewer #2: The study aims to clarify the prediction error of plastid and mitochondrion proteomes across the plant kingdom with evolutionary distance from the training standard and speculate ways to improve prediction accuracy/precision. It is a well written manuscript. While this is an ongoing issue of high relevance the novelty of the study is limited and concrete steps for improvements are not given.

Overall, the claim that the proteomes predicted and experimental vary due to errors and differences in plant physiology across evolution are known. Secondly, the prediction algorithms are mostly trained on e.g. Arabidopsis and will work best on this species as there has been the most data to train. So this is not a novel insight.

Abstract is not clearly stating that the study is limited to plastid and mitochondria and which 4 species were used to assess the predictors. The authors make claims using only Arabidopsis, Chlamydomonas, Zea mays, P patens to choose the best predictor. There are a lot more experimental data out there so the authors could do a more substantial analysis across several individuals computationally and then also using experimental data sets. However, the authors need to compare their assessment of 3 predictors with other assessments. For example, the comparison of >10 predictors against multiple experimental data sets and classifiers in Arabidopsis (e.g. SUBAcon and/or the SUBA4 (Bioinformatics. 2014 Dec 1;30(23):3356-64. doi: 10.1093/bioinformatics/btu550. Epub 2014 Aug 22.). The latter had a full assessment of ~20 predictors including TargetP and WolfPSORT. This paper also found a TargetP to be one of the strongest predictors for plastid using a far more rigid methodology for comparing to experimental data alone. The authors may also need to consider Predotar (Proteomics. 2004 Jun;4(6):1581-90. doi: 10.1002/pmic.200300776.) which is still a strong predictor for mitochondria using NTS and the training set may be more widespread perhaps.

Secondly, the cropPAL dataset mentioned by the authors has 12 species including experimental data and was used to describe the divergence of the subcellular proteomes to some degree across 6 monocots and 6 dicots from different branches of the plant kingdom ( Plant J. 2020 Nov;104(3):812-827. doi: 10.1111/tpj.14961. Epub 2020 Sep 16). Can the authors discuss this and emphasis how their claim is different or more substantial?

Also, the experimental sets used to verify predictors need to be described in more detail. What were the QC parameters in this data set. The authors assume that these are correct and maybe complete. Experimental data sets are also error prone to contamination and missing proteins. If using MS data, this method tends to detect similar protein families across species, namely prevalent proteins and those that form detectable peptides. Hence all proteomes generated by MS tend to plateau towards the same group of but always miss the rest of proteins. In that light, the authors state a large number of false positives but to what degree is this correct? The experimental plastid/mitochondria datasets are much smaller than expected so a lot of the false positives may actually be correct? Another point not discussed by the authors is how experimental discrepancies are being dealt with?

Most literature expect plastid and mitochondrion proteomes to be around 10-15%. The authors mention 5%. The evidence for this needs to be described.

The evolutionary approach is very interesting and more convincing and as mentioned earlier I would not readily assume that the differences seen in the prediction are errors. Can the authors clarify that the prediction error definition is? Is it prediction compared to experimental? If so, what about missing experimental proteins that are in the plastid/mitochondrion?

Proteins are known to be conserved as well as in distinctly different organelles amongst different species. The authors show the training set for TargetP per species but what is the distribution of the training standard compared to the test data sets of 147 species? Is it comparable or is the training standard skewed differently to the test? This needs to be clearer in the text.

There are several individual predictors and ensemble predictors/classifiers that take evolution into account. These have not been considered or discussed (e.g. BaCelLo - Bioinformatics. 2006 Jul 15;22(14):e408-16. doi: 10.1093/bioinformatics/btl222.). How do more balanced predictors perform?

The results suggest also that apart from distance there are other relationships limited to e.g. clades etc. Which is interesting and may warrant more investigation as this has not been done systematically before. The authors suggest an algorithm with better adjustment to evolutionary distance but do not propose how this would work.

Dual targeted proteins: This has been looked at a few times (e.g. in the cropPAL data set and in Arabidopsis versus Zea mays or Oryza and other species to some degree) and should be discussed here. In general, most predictors struggle predicting dual targeted proteins even if training on the species they are predicting. This is a long-standing issue and yet has not been well addressed in this study! The authors need to discuss the literature (e.g. New Phytol. 2009;183(1):224-236. doi: 10.1111/j.1469-8137.2009.02832.x.)

Choosing protein families is an interesting approach as such and it leads to the conundrum to include families with proteins only existent in the plastid or in the mitochondria versus families with proteins that have members sent to different locations. How are these considered? This may have introduced a lot of missed proteins that are not really missed.

Pan proteome and genome training would be good idea but how to is not described.

Other comments:

Figures need stand-alone legends. Please review each and add missing explanations particularly the supplementary.

Reviewer #3: The paper titled "Performance of localization prediction algorithms decreases rapidly with the evolutionary distance to the training set increasing" presents a very interesting and systematic evaluation to plant organelle proteomes. The study took a big collection of known plant organelle proteome data as a basis to evaluate three major computational prediction tools. I think this is a good study. I also personally favor its conclusion. I have only two minor comments, as follows:

1 As this is computational biology, I think the authors should present the major mathematical formulations in the main text, not only in supps.

2 I think the code and datasets for reproducing the study should also be deposited publicly. Zenodo and Github are recommended.

**Have the authors made all data and (if applicable) computational code underlying the findings in their manuscript fully available?**

Reviewer #1: Yes

Reviewer #2: Yes

Reviewer #3: **No: **Please deposit codes publicly, in GitHub.

PLOS authors have the option to publish the peer review history of their article (what does this mean?). If published, this will include your full peer review and any attached files.

Reviewer #1: **Yes: **Kenta Nakai

Reviewer #2: No

Reviewer #3: No
---

## [Decision Letter · Decision Letter 1]

17 Sep 2024

Dear Dr. Gould,

Thank you very much for submitting your manuscript "Reliability of plastid and mitochondrial localisation prediction declines rapidly with the evolutionary distance to the training set increasing" for consideration at PLOS Computational Biology.

As with all papers reviewed by the journal, your manuscript was reviewed by members of the editorial board and by several independent reviewers. In light of the reviews (below this email), we would like to invite the resubmission of a significantly-revised version that takes into account the reviewers' comments.

You will note that Reviewer 3, in particular, has important outstanding concerns that need to be considered and addressed in a revised paper.

We cannot make any decision about publication until we have seen the revised manuscript and your response to the reviewers' comments. Your revised manuscript is also likely to be sent to reviewers for further evaluation.

Sincerely,

Daniel A Beard

Section Editor

PLOS Computational Biology

Daniel Beard

Section Editor

PLOS Computational Biology

Reviewer's Responses to Questions

**Comments to the Authors:**

Reviewer #1: Since I confirmed that the authors' responses to my points are reasonable, I now recommend its acceptance as is.

Reviewer #2: The authors have put substantial work into improving the manuscript. It is now much clearer that this study focusses on plastid and mitochondria across cornerstones of the plant kingdom.

I agree that many prediction algorithms are not accessible 5 or more years after publication, highlighting a serious issue in our system.

The authors also adjusted and increased their conclusions substantially to reflect the data limitations and findings.

Reviewer #3: I do not think the authors face my questions and concerns with a consolidated scientific response. I believe none of my concerns is addressed. In particular, no reproducible instruction is provided in zenodo. The codes and XLS files are thus useless. Therefore, I am sorry that I suggest to reject this paper because the background information (data/codes/instructions for reproducing) is not fully published.

**Have the authors made all data and (if applicable) computational code underlying the findings in their manuscript fully available?**

Reviewer #1: None

Reviewer #2: Yes

Reviewer #3: **No: **No reproducible instruction is provided based only on public information

PLOS authors have the option to publish the peer review history of their article (what does this mean?). If published, this will include your full peer review and any attached files.

Reviewer #1: **Yes: **Kenta Nakai

Reviewer #2: No

Reviewer #3: No
---

## [Editor Report · Decision Letter 2]

17 Oct 2024

Dear Dr. Gould,

We are pleased to inform you that your manuscript 'Reliability of plastid and mitochondrial localisation prediction declines rapidly with the evolutionary distance to the training set increasing' has been provisionally accepted for publication in PLOS Computational Biology.

Best regards,

Anders Wallqvist

Academic Editor

PLOS Computational Biology

Daniel Beard

Section Editor

PLOS Computational Biology

---

## [Editor Report · Acceptance letter]

6 Nov 2024

PCOMPBIOL-D-24-00460R2 

Reliability of plastid and mitochondrial localisation prediction declines rapidly with the evolutionary distance to the training set increasing

Dear Dr Gould,

I am pleased to inform you that your manuscript has been formally accepted for publication in PLOS Computational Biology. Your manuscript is now with our production department and you will be notified of the publication date in due course.

With kind regards,

Anita Estes
